# Design, Synthesis and Antibacterial Activity of Coumarin-1,2,3-triazole Hybrids Obtained from Natural Furocoumarin Peucedanin

**DOI:** 10.3390/molecules24112126

**Published:** 2019-06-05

**Authors:** Alla V. Lipeeva, Danila O. Zakharov, Liubov G. Burova, Tatyana S. Frolova, Dmitry S. Baev, Ilia V. Shirokikh, Alexander N. Evstropov, Olga I. Sinitsyna, Tatyana G. Tolsikova, Elvira E. Shults

**Affiliations:** 1Novosibirsk Institute of Organic Chemistry, Siberian Branch of the Russian Academy of Sciences, Lavrentyev Ave. 9, 630090 Novosibirsk, Russia; mond_05@list.ru (A.V.L.); zakharov@nioch.nsc.ru (D.O.Z.); mitja2001@gmail.com (D.S.B.); TG_tolstikova@mail.ru (T.G.T.); 2Department of Microbiology, Immunology and Virology, Novosibirsk State Medical University, Krasny Prospect 52, 630091 Novosibirsk, Russia; mic.bio.lgb@mail.ru (L.G.B.); ishirok@gmail.com (I.V.S.); mic.bio.ave@mail.ru (A.N.E.); 3The Federal Research Center Institute of Cytology and Genetics, Acad. Lavrentyev Ave. 10, 630090 Novosibirsk, Russia; frolova@bionet.nsc.ru (T.S.F.); oisinits@bionet.nsc.ru (O.I.S.); 4Novosibirsk State University, Pirogova Str. 1, 630090 Novosibirsk, Russia

**Keywords:** coumarin, furocoumarin, Sonogashira coupling, CuAAC reaction, antibacterial activity

## Abstract

Synthesis of 1,2,3-triazole-substituted coumarins and also 1,2,3-triazolyl or 1,2,3-triazolylalk-1-inyl-linked coumarin-2,3-furocoumarin hybrids was performed by employing the cross-coupling and copper catalyzed azide-alkyne cycloaddition reaction approaches. The synthesized compounds were evaluated for their in vitro antibacterial activity against *Staphylococcus*
*aureus*, *Bacillius*
*subtilis*, *Actinomyces*
*viscosus* and *Escherichia*
*coli* bacterial strains. Coumarin-benzoic acid hybrids **4с**, **42с** and 3-((4-acetylamino-3-(methoxycarbonyl)phenyl)ethynyl)coumarin (**29**) showed promising activity against *S*. *aureus* strains, and the 1,2,3-triazolyloct-1-inyl linked coumarin-2,3-furocoumarin hybrid **37c** was endowed with high selectivity against *B. subtilis* and *E. coli* species. The in vitro antibacterial activity of **4с**, **29**, **37c** and **42с** can potentially be compared with that of a number of modern antibiotic drugs used in the clinic, suggesting promising prospects for further research. A detailed study of the molecular interactions with the targeted protein MurB was performed using docking simulations and the obtained results are quite promising.

## 1. Introduction

As stated in a WHO report, in the last few decades, the incidence of microbial infections has increased dramatically together with emergence of antimicrobial-resistant strains [1]. Increasing instances of antimicrobial drug resistance requires the design and synthesis of new small molecules with higher affinity and specificity for their potential targets to serve as antibiotics. Coumarins, naturally plant-derived compounds with a benzopyrone moiety, possess a wide variety of biological activities. Series of coumarin derivatives are being extensively studied due to their broad array of biological activities, low toxicity, and lower drug resistance properties [2].

Various naturally-isolated coumarins, as well as their chemically modified analogs, are active against numerous bacterial strains, including those which have developed multidrug resistance [2]. Among these compounds of interest are the coumarin-1,2,3-triazole hybrids (Figure 1) [3,4]. 1,2,3-Triazoles have been nuclei of choice in recent years because of their excellent favorable safety profile, latent ability to form hydrogen bonds, moderate dipole character, rigidity and stability under in vivo conditions and capability of interacting with biomolecular targets [5,6]. A set of coumarin hybrids **A** (Figure 1) having a 1,2,3-triazole moiety in the C-7 position on the coumarin core was assessed for their in vitro antimicrobial activities against Gram-positive and Gram-negative pathogens [7]. The obtained results showed that all hybrids of type **A** displayed considerable activity against the tested strains, and SAR studies revealed that the substituent and the length of alkyl spacers in the 1 and 4 position of the triazole ring have profound effects on the antimicrobial potency. Morpholinylmethyl- and piperazinylmethyl (N-R_1_,R_2_)-containing hybrids exhibited noticeable activity against various bacterial pathogens and were more potent than the hybrids incorporating a phthalimidomethyl moiety in the 4 position of the triazole ring. The 6,8-disubstituted coumarin-1,2,3-triazole conjugates **B** (Figure 1) exhibited significant in vitro antibacterial activity [8,9]. Investigation of the antimicrobial activity of this type of hybrid compounds led to the identification of several different structural frameworks. The obtained results suggested that both the length of the spacer (n) and also the nature of the substituent (H, OH, OMe) in the 6 and 8 positions modified the lipophilicity of the hybrids, and this in turn affected the antibacterial activity. In these molecules, the triazole rings are linked with the coumarins by a methylene oxygen. The antibacterial activity was correlated with the length of the spacer (n), and the contribution order of substituents was the following: H > OMe > OH.

The substituted coumarins with 1,2,3-triazole substituent in the 3 position (compounds **C**) exhibited activity which was dependent on the nature of the substituent on the triazole ring (alkylamide, alkylpeptide) [10].

All coumarin-1,2,3-triazole hybrids of the type **D** (Figure 1) displayed promising antibacterial and antifungal properties, which were comparable with the reference drug griseofulvin [11]. The SAR revealed that the substituent at C-5, C-6, and/or C-8 position of coumarin moiety have a great influence on the antimicrobial activity, and 6-Br was favorable for activity against the bacterial strain *E*. *coli* and the fungus *A*. *niger*, while a 6-Me benefitted the activity against *B*. *subtilis*; monohalo hybrids were more potent than their bis-halo counterparts (in the 6 and 8 positions) against all strains.

Thus, the design and synthesis of novel coumarinotriazole derivatives offers a prospective route for accessing new molecules with improved antibacterial activity profiles. Our research interests include the design of convenient ways to access polysubstituted natural furocoumarins [12,13,14,15,16] and coumarins [17,18] and the assessment of their potential as bioactive agents. Herein we report the synthesis of a range of coumarins containing a triazole substituent in the 3 or 6 position of the coumarin core. Our attention was concentrated on the synthesis of compounds bearing both a previously identified pharmacophoric moiety and a substituted triazole ring. Thus, we describe the synthesis of a range of mixed compounds having coumarin-2,3-dihydrofurocoumarin hybrid structures linked through an 1*H*-1,2,3-triazole ring or alkyne-methylene-triazolyl bridge at the C2-C3′ atoms. As starting compounds we used the natural linear furocoumarin peucedanin (1) and the coumarin peuruthenicin (officinalin, 7-hydroxy-6-(methoxycarbonyl)coumarin, 2, Scheme 1). The Pd-catalyzed coupling methods and Cu-catalyzed azide-alkyne cycloaddition (CuAAC reaction) were the main routes of synthesis. The antimicrobial activity of the triazolyl-substituted coumarins against a panel of clinically relevant bacterial strains were also investigated and discussed.

## 2. Results and Discussion

### 2.1. Chemical Synthesis

The umbelliferon carboxylic acid **3**, the starting compound for the synthesis of triazolyl substituted coumarins **4a**–**c** (Scheme 1), was conveniently obtained by treatment of peuruthenicin (**2**) with NaOH in dioxane [19]. The peptide coupling reaction of **3** with propargyl amine **5** using DCC as a coupling reagent gave the corresponding 6-(prop-2-ynylcarbamoyl)umbelliferone (**6**) in 67% yield. We have chosen the most simple and well-characterized variant, namely, carrying out the reaction in a CH_2_Cl_2_-water mixture, with catalysis of the Cu(I) ions generated in situ from CuSO_4_ and sodium ascorbate [12]. CuAAC-reaction of ethynyl coumarin **6** with three positional azidobenzoic acid isomers **7a**–**с** led to the 6-[1-(carboxyphenyl)-1,2,3-triazol-4-yl]carbamoyl)- umbelliferones **4a**–**c** in 50–62% yields.

The synthetic route followed for the synthesis of the desired novel coumarins substituted with a triazole ring in the C-3 position **8**–**10** is outlined in Scheme 2. Bromination of peuruthenicin (**2**) with dioxane dibromide (2.2 eq) in CH_2_Cl_2_ proceeds selectively and led to the formation of 3-bromopeuruthenicin (**11**) as the sole product in 88% isolated yield. 3-Аzidocoumarin **12** was conveniently obtained in 68% yield, by reacting the corresponding 3-bromopeuruthenicin (**11**) with sodium azide in DMF. The second alkyne building block was obtained from amino acids in two steps. At the first step, 9-aminopelargonic acid, d,l-2-aminobutyric acid or L-phenylalanine hydrochloride were transformed to subsequent methyl ethers of the amino acid hydrochloride in nearly quantitative yield requiring no further purification. Reaction of amino acids methyl esters hydrochloride with propargyl bromide in the presence of K_2_CO_3_ in dry DMF gave alkynes **13**–**15**, which were purified through the column chromatography on silica gel using a CHCl_3_/EtOH mixture as an eluent.

The interaction of the new azide **12** with methyl 9-(prop-2-ynylamino)nonanoate **13** under above catalytic conditions was carried out at 20 °C for 4 h (monitoring by TLC) and led to the target compound **8** in the isolated yield 67%. The reaction of azide **12** with terminal alkynes **14**, **15**, proceeds with the formation of substituted coumarins **9** or **10** in high yield by performing this reaction in the presence of CuI and Et_3_N (method *d*). By using of CuSO_4_ and sodium ascorbate as the catalysts in the reaction of **12** with **15**, the isolated yield of coumarin **10** was decreased to 34%.

Thus we obtained 3-(1*H*-1,2,3-triazol-1-yl)peuruthenicin derivatives **8**–**10**. It was also important for us to identify the influence of the position of coumarin as the substituent in the C-4 or C-1 of the triazole ring. To obtain 3-(1*H*-1,2,3-triazol-4-yl)peuruthenicins we were interested in the synthesis of 3-(ethynyl)peuruthenicin (**16**, Scheme 3).

For this purpose we studied the activity of 3-bromopeuruthenicin (**11**) in the Sonogashira cross-coupling reaction. Besides providing data about the reactivity of 3-bromopeuruthenicin (**11**) in the cross-coupling reaction with various terminal alkynes, this study served to make available a larger panel of peuruthenicin derivatives for testing their antimicrobial activity. Therefore, the reaction between 3-bromopeurutenicine (**11**) with phenyl acetylene (**17**) was optimized first, attempting to obtain the cross-coupling derivative **18** under different conditions. After considerable experimentation, we found that the cross-coupling proceeds in a benzene solution in the presence of a catalytic amounts of *trans*-dichlorobis(triphenylphosphine)palladium(II), copper(I) iodide, and Et_3_N as a base (conditions a, Scheme 3) and led to the corresponding 3-(phenylethynyl)coumarin (**18**) in 65%yield after column chromatography on silica gel. Using Bu_4_NBr as an additive, DMF as a solvent and Et_3_N as a base did not improve the yield of compound **18** (conditions b). On the other hand, using K_2_CO_3_ as a base in benzene or DMF at 80–100 °C also met with failure, and the reaction afforded yields of 45% or 42%, respectively, and a yield of 20% was recorded when the transformation was performed using (*i*Pr)_2_NH as a base. With the set of optimum conditions in hands, we found that the reaction of coumarin **11** with aryl(hetaryl)acetylenes **19**–**24** proceeds with the formation of compounds **25**–**30** (52–68% yields. A perspective route for synthesis of terminal alkynes (for example, 3-ethynylpeuruthenicin (**16**) was the Sonogashira coupling of **11** with accessible 2-methylbut-3-yn-2-ol (**31**) and subsequent elimination of acetone from the 3-alkynylcoumarin **32**. We found that the reaction of **11** with alkyne **31** (conditions *a*) led to the formation of 3-(3-hydroxy-3-methylbut-1-ynyl)coumarin (**32**, (66% yield). However, the transformation of the alcohol **32** into the terminal alkyne using the known conditions [20] was unsuccessful. The copper and palladium-catalyzed Sonogashira coupling of 3-bromopeuruthenicin **11** with trimethylsilylacetylene (**33**) (using conditions b) rqwuired a long reaction time and proceeds with the formation of traces of 3-(trimethylsilyl- ethynyl)peuruthenicin (**34**). Those results forced us improve this reaction. Currently, microwave-assisted organic synthesis (MAOS) is receiving increasing attention as a valuable alternative to the conventional heating to speed up chemical reactions, and microwave irradiation has been applied to Sonogashira coupling reactions [21,22]. It should be noted that MAOS has significant advantages that include simplicity in operation, increased reaction rates, and improved reaction yields. Performing the microwave-assisted reaction of 3-bromopeuruthenicin (**11**) and trimethylsilylacetylene (**33**) using Pd(PPh_3_)_2_Cl_2_, CuI as catalysts and Et_3_N as a base in toluene solution under microwave irradiation at 100 °C for 2 h give the cross-coupling product **34** in 64% isolated yield. Thus, the use of microwaves was found to significantly improve the reaction yield and shorten the reaction time. Desilylation of compound **34** using CsF in MeOH in the presence of benzyltriethylammonium chloride (TEBA) afforded 3-ethynylpeuruthenicin (**16**, 85% yield) which was purified by column chromatography on silica gel using CHCl_3_ as an eluent.

At the next step of our study, the acetylene building block **16** was reacted with 2-azidooreozelone (**35**) [14]. Carrying out the reaction in a СH_2_Сl_2_-water mixture, with catalysis of the Cu(I) ions generated in situ from CuSO_4_ and sodium ascorbate we obtained the coumarin-2,3-dihydrofurocoumarin hybrid linked through an 1*H*-1,2,3-triazole ring **36** isolated in 70% yield (Scheme 4).

We also performed the synthesis of coumarin-2,3-dihydrofurocoumarin hybrids **37a**–**c**, containing a triazolylmethylene-1-inyl linker group (Scheme 4). The synthetic route involved the Sonogashira’s reaction of new 2-ethynylalkyltriazolyl substituted furocoumarins **38a**–**c** with 3-bromopeuruthenicin (**11**, Scheme 4). Compounds **38a**–**c** (yield 62–74%) were prepared by the CuAAC reaction of 2-azidooreoselone (**35**), with 1,6-heptadiyne (**39**), 1,7-octadiyne (**40**) or 1,9-decadiyne (**41**). Reaction of 3-bromopeuruthenicin (**16**) with alkynes **38а**–**с** in the Sonogashira reaction conditions afforded the coumarin-2,3-dihydrofurocoumarin hybrids linked through 1*H*-1,2,3-triazolylalk-1-inyl groups **37a**–**c** (52–57% yields).

### 2.2. Biological Screening

The new coumarins **4a**–**c**, **11**, **29**, **30**, coumarin-dihydrofurocoumarin hybrids **36**, **37a**–**c**, and the 7-triazolylsubstituted coumarins **42a**–**c** (Figure 2), the synthesis of which was described in our previous paper [23] were screened for their in vitro antibacterial activity against Gram positive bacterial strain *Staphylococcus aureus* 209 ATCC 6538P. The antibacterial activity was studied by serial dilution in a liquid nutrient medium [24]. The minimum inhibition concentrations (MIC) of the test cultures were determined. The most promising substances were those with indicators of MIC = 100 µg/mL and less. Table 1 reveals that coumarino-1,2,3-triazole derivatives **4c** and **42c** and 3-arylethynylpeuruthenicin **29** showed excellent antibacterial activity, with MIC values ranging from 0.16–0.41 μg/mL against the tested microorganism and they prove to be better than that of the reference drugs ceftriaxone (0.97 µg/mL) and streptomycin (1.89 µg/mL).

In the series of coumarin-benzoic acid hybrids **4a**–**c**, **42a**–**c** in view of the position of the carboxylic group in the benzoic acid moiety, we considered that compounds containing this substituent in the 4 position **4c**, **42c** were more active than compounds **4a**,**b** and **42a**,**b**, respectively, so the carboxylic acid group is an excellent microbial ligand which can bind more effectively at the active site of receptor. The difference in antibacterial activity profile of 3-bromocoumarin **11** and 3-(arylethynyl) coumarins **26**, **29**, **30** indicates the importance of the nature of the functional substituent on the triple bond in the C-3 position of coumarin. Thus, compound **29** containing an anthranilic acid methyl ester fragment on the 3-ethynylcoumarin nucleus showed greater activity than compounds with 4-tolyl- or pyridine substituents at the triple bond in 3-ethynylcoumarin. It should be noted that the biological effect of anthranilic acid derivatives is based on their ability to act as modulators of the nuclear peroxisome proliferator activated receptors (PRAR) and the farnesoid X (FXR) receptors, which fulfill crucial roles in metabolic balance [25,26]. In this direction the antibacterial activity of compounds **37a**–**c**, having a methylene-triazolyl-furocoumarin substituent on the triple bond of 3-ethynylcoumarin was dependent on the length of the C-methylene linker. The activity of the most active hybrid compound **37c** is due the presence of a hexamethylene-1-inyl linker group. Coumarin-2,3-dihydrofurocoumarin hybrid **37c** was found to exhibit good potency at 51.25 mg/mL of MIC, while the parent compounds **1** and **2** exhibited a lack of activity (Table 1).

The results of study of promising substances **4с**, **29**, **37c** and **42с** on other strains of *Staphylococcus aureus* and *Actinomyces viscosus* U-18 are presented in Table 2. Further study on the *S. aureus* strain confirmed the high activity of compounds **4с**, **29** and **42c**. Compound **4c** (carboxamidotriazolyl- benzoic acid substitution at the C-6 position of the coumarin core) showed good activity against *A*. *viscosus* compared with compound **42c** which showed moderate activity against *A. viscosus*. Characteristically, that compound **42c** with a triazolylbenzoic acid substituent in the C-7 position possessed the highest activity against *S*. *aureus* “Viotko” bacterial strains. Of interest was also the high antibacterial activity of 3-ethynylcoumarin with methylanthranilate substituent **29** on the all tested *S. aureus* strains. This compound will be further used as the scaffold for structural optimization to develop more potent and selective antibacterial agents.

Next set of experiments was dedicated to the analysis of the antibacterial potential of coumarin-2,3-dihydrofurocoumarin hybrids **36**, **37a**–**c** (Table 3). The in vitro antibacterial activity of those compounds was additionally tested against *Bacillius subtilis* and *Escherichia coli* (JM 109) bacterial strains. The obtained results were compared with the known tumorogenic compound 4-nitroquinolin-1-oxide (NQO) and presented as an average concentration of inhibitory 50% bacterial proliferation (incubation time 20 h). It can be observed that from the series of coumarin-2,3-dihydrofurocoumarin hybrids only dimeric compound **37c** with the 1,2,3-triazolyloct-1-inyl linker group displayed promising activity against *B. subtilis* and also *E. coli* (JM 109) bacterial strains. On the contrary to this, variously 1,2,3-triazolyl or 1,2,3-triazolylhex-1-inyl or 1,2,3-triazolylpent-1-inyl linked hybrids **36**, **37a**,**b** were deprived of anti-bacterial activities (MICs were >1000 mg/mL).

#### Molecular Docking

Molecular docking protocol is an essential tool to mimic the natural course of interaction of the ligand with active sites of receptors through lowest binding energies in which two molecules fit together in 3D space. Here, we discuss the mechanism of interaction between coumarinotriazole derivatives **3, 4** and **5** with the MurB protein (PDB ID: 1HSK). MurB is an attractive target protein in bacterial infection diseases and it is an essential enzyme which takes part in amino sugar metabolism which reduces the enolpyruvyluridine diphosphate N-acetyl glucosamine (EP-UNAG) as an intermediate in the assembly of the UNAM-pentapeptide (m-A2pm) protein to uridine diphosphate N-acetylMuamic acid (UNAM) of cell wall precursor [27]. For the calculations, the XRD model of *S. aureus* N-acetylenolpyruvylglucosamine reductase (MurB) with PDB ID 1HSK was chosen (resolution 2.3 Å) [28]. To model a possible mechanism of MurB inhibition, molecular docking of new coumarins was performed at the binding site of flavin adenine dinucleotide (FAD) in the Glide application [29]. We have screened the coumarinotriazoles **4a**–**c, 8, 9,** and **42a**–**c** and also coumarin-furocoumarin hybrids **36**, **37b**, and **37c**. The molecules **4a**–**c, 37c** and **42a,c** strongly approach the MurB protein receptor as shown by their minimum binding energies −8.416–−8.983 Kcal/mol (Table 4). The docking results were found to be in good agreement with in vitro antibacterial experimental MIC values (Table 1, Table 2 and Table 3). The FAD binding site is saturated with polar amino acids. This facilitates formation of a large number of hydrogen bonds due to the large number of polar groups in the FAD molecule. Additional stabilization in the binding site provided stacking interactions of aromatic cycles of adenine. Inspection of the binding mode demonstrated, that compounds **4c**, **37c** and **42c** successfully combine in their structure a large number of polar groups and π-systems (Figure 3). The presence of the carboxy function in compounds **4c**, **42c** allowed the formation of hydrogen bond with amino acid ARG225 residue (Figure 3A–C). As shown in Figure 3C, the inhibitor **37c** formed addition hydrogen bonding interactions with the active site residues: the furan ring C=O with SER 238 (2.92 Å) and the coumarin ring C=O with TYR 155. All those interactions contribute to a spatial orientation close to FAD and the formation of hydrogen bonds with the same amino acid residues as FAD (Figure 3D). We found that the MurB protein receptor amino acids ASN80, SER82, SER238, GLY81 are the most active sites responsible for interactions with the ligand. Important interaction centers at the binding site are arginines 225 and 310, protein chain section 79–83, glycines 145, 146 and 153, isoleucine 140 and proline 141, valine 199. New coumarins interact with almost all these centers, except for 140–141, and also form bonds characteristic only for these molecules. The formation of interactions with aromatic amino acid residues increases the stability of the conformations of new coumarins at the binding site.

## 3. Conclusions

In summary, we have synthesized new series of coumarinotriazole compounds using [2 + 3]-cycloaddition reactions or Sonogashira cross-coupling reactions as the main approaches. The synthesized coumarino-triazole type derivatives were screened for their in vitro antimicrobial activity. Compounds **4c** and **42c**, having the 4-(carboxyphenyl)triazolyl substituent in the 6 or 7 position of the coumarin ring showed excellent antibacterial activity against *S. aureus* strains, with MIC values of 0.16–3.75 μg/mL and 0.21–6.28 μg/mL respectively. Most of the compounds of series **11**, **29** and **30** with bromine, and aryl(hetaryl)substituents in the 3 position revealed significant MIC values (between 0.41 to 2.0 μg/mL for compound **29**), indicating that 3-substituted coumarin analogues show promising antibacterial activity and are key compounds for further development as antibacterial agents. Coumarin-2,3-dihydrofurocoumarin hybrid compound **37c** was found to be selective against *Bacillius subtilis* and *E. coli*, with MIC values of 0.02–0.15 μg/mL. A molecular docking study was performed for the most active compounds against the MurB protein. Molecular docking results were well corroborated with the in vitro antibacterial activity findings. Finally, the successful synthesis and antimicrobial evaluation along with docking study of new coumarino-triazole scaffolds obtained on the base of accessible plant coumarins peucedanin and peuruthenicin will provide further advantages to design and develop triazole derivatives with selective antibacterial activity.

## 4. Experimental Section

### 4.1. General Information

NMR spectra were acquired on Bruker AV-400 (^1^H: 400.13 MHz, ^13^C: 100.78 MHz) or Bruker AV-600 (^1^H: 600.30 MHz, ^13^C: 150.95 MHz) instruments (Bruker BioSpin GmbH, Rheinstetten, Germany), using tetramethylsilane (TMS) as an internal standart. In the description of the ^1^H- and ^13^C-NMR spectra, the furocoumarin and coumarin skeleton atoms numeration system given in structures **1** and **2** was used. Chemical shifts are reported in parts per million (ppm). The IR spectra were recorded by means of the KBr pellet technique on a Bruker Vector-22 spectrometer. The UV spectrum were obtained on an HP 8453 UV Vis spectrometer (Hewlett-Packard, Waldbronn, Germany). HRMS spectra were recorded on a DFS mass spectrometer (Thermo Fisher Scientific, Waltham, MA, USA), evaporator temperature 180–220 °C, EI ionization at 70 eV). The specific rotation values [α]_D_ were determined on a PolAAr 3005 polarimeter (Rudolph Research Analytical, Hackettstown, NJ, USA), and expressed in (deg·mL)/(g·dm), while concentration was expressed in g per 100 mL of solution. Melting points were determined using Stuart SMP30 melting point apparatus (Bibby Scientific, Staffordshire, UK). The microwave irradiation reaction was performed in a Microwave 50 reactor (Anton Paar, Graz, Austria). Elemental analysis was carried out on an 1106 Elemental analysis instrument (Carlo-Erba, Milan, Italy). The molecular weights of compounds **37a**–**c**, **38c** were determined using a Knauer vapor pressure osmometer (Knauer, Berlin, Germany). Spectral and analytical investigations were carried out at Collective Chemical Service center of Siberian Branch of the Russian Academy of Sciences.

Reaction products were isolated by column chromatography on silica gel 60 (0.063–0.200 mm, Merck KGaA, Darmstadt, Germany) eluting with chloroform and chloroform-ethanol (100:1; to 25:l). The reaction progress and the purity of the obtained compounds were monitored by TLC on Silufol UV-254 plates (CHCl_3_-EtOH, 9:1; detection under UV light or by treatment with iodine vapor).

The chemicals used: dioxane dibromide, trifluoromethanesulfonic anhydride, NaN_3_, CuSO_4_, sodium ascorbate, CuSO_4_, CuI, N,N-dicyclohexylcarbodiimide (DCC), propargyl amine hydrochloride (**5**), propargyl bromide (80% in toluene solution), aryl acetylenes **17**, **19**–**2**, **24**, 2-methylbut-3-yn-2-ol (**31**), trimethylsilylacetylene (**33**), 1,6-heptadiyne (**39**), 1,7-octadiyne (**40**) and 1,9-decadiyne (**41**) were purchased from Aldrich (St. Louis, MO, USA) or Alfa Aesar (GmbH, Karlsruhe, Germany). Dichlorobis(triphenylphosphine)palladium(II) was obtained as described in [30]. Peucedanin (**1**) was isolated from *Peucedanum morisonii* according to [31]. Peuruthenicin (**2**) [32], umbelliferone carboxylic acid **3** [19], azidobenzoic acids **7а**–**с** [33], methyl ether of 9-aminopelargonic acid hydrochloride [34], D,L-2-aminobutyric acid and L-phenylalanine hydrochlorides [35], methyl 2-(prop-2-ynylamino)butanoate (**14**), and (*S*)-methyl 3-phenyl-2-(prop-2-ynylamino)propanoate (**15**) [36], methyl ethynylantranilate (**23**) [37] and 2-azidooreoselone (**35**) [14] were synthesized according to published procedures. Solvents (CH_2_Cl_2_, CHCl_3_, MeCN, DMF, THF, benzene, dioxane) and Et_3_N were purified by standard methods and distilled under a stream of argon just before use. Copies of NMR spectra (^1^H & ^13^C) are given in Appendix A.

### 4.2. Syntheses and Spectral Data

*7-Hydroxy-2-oxo-N-(prop-2-ynyl)-2H-chromene-6-carboxamide* (**6**) To a stirred solution of umbelliferone-6-carboxylic acid (**3**, 500 mg, 2 mmol) and hydroxybenzotriazole (135 mg, 1 mmol) in DMF (5 mL) was added propargylamine hydrochloride (**5**, 120 mg, 2.1 mmol) and Et_3_N (0.27 mL, 2 mmol) and the reaction mixture was left for 5 min. Then diisopropylcarbodiimide (300 mg, 2.4 mmol) was added and the mixture was stirred for 2 days. The reaction mixture was poured into a Petri dish to evaporate in air, the residue was dissolved in CH_2_Cl_2_ (10 mL) and acetone was added (5 mL). The precipitate was filtered, the solution was evaporated and residue was dried. The product was isolated by fractional precipitation from acetone. Compound **6** was obtained in 67% yield (320 mg). Colorless powder. М.р. 188 °С (dec.). IR (KBr, *ν*, cm^−1^): 3388, 3286, 3243, 3059, 2921, 2852, 2100, 1737, 1708, 1649, 1620, 1598, 1570, 1500, 1392, 1344, 1310, 1234, 1213, 1144, 911, 854, 837, 735, 688, 639, 628. UV (EtOH) λ_max_, (lgε): 222(3.22), 240 (3.35), 329 (2.89), 372 (2.88) nm. ^1^H-NMR (CDCl_3_ + CD_3_OD, 400 MHz, δ_H_): 3.16 (s, 1Н, ≡СН), 4.20 (s, 2Н, СН_2_), 6.26 (d, *J* = 9.6 Hz, 1Н, Н-4), 6.83 (s, 1Н, Н-8), 7.76 (d, *J* = 9.6 Hz, 1Н, Н-3), 8.10 (s, 1Н, Н-5). ^13^C-NMR (CDCl_3_+CD_3_OD, 100 MHz, δ_C_): 45.8, 71.0, 78.8, 103.9, 111.4, 112.6, 113.8, 129.3, 144.0, 157.1, 161.2, 162.4, 167.1 (С=O). Anal. calcd for C_13_H_9_NO_4_: C, 64.20; H, 3.73; N, 5.76; found: С 64.33; Н 3.77; N 5.65.

#### 4.2.1. General Method for the Synthesis of Compounds **4a**–**c**

A solution of the appropriate azidobenzoic acid **7a**–**с** (35 mg, 0.14 mmol) in CH_2_Cl_2_ (10 mL) was mixed with a solution of sodium ascorbate (42 mg, 15 mol%, 0.021 mmol) and СuSO_4_·5H_2_O (1.2 mg, 5 mol%, 0.007 mmol) in Н_2_О (5 mL) and 6-(N-propynyl)carboxamidocoumarin (**6**, 20 mg, 0.14 mmol) was added while stirring. The reaction mixture was stirred for 3 h at room temperature and additional for 1 h at 40 °С. After the completion of the reaction, the mixture was diluted with Н_2_О (10 mL), the product was extracted with CH_2_Cl_2_ (4 × 10 mL), the combined extracts were dried over anhydrous MgSO_4_, and the solvent was removed at reduced pressure. The product was purified by column chromatography (eluent CHCl_3_).

*2-(4-((7-Hydroxy-2-oxo-2H-chromene-6-carboxamido)methyl)-1H-1,2,3-triazol-1-yl)benzoic acid* (**4a**). Colorless powder, m.p. 201–202 °C (ethanol). Yield 50%. IR (KBr, *ν*, cm^−1^): 3367, 3257, 3066, 2922, 2852, 2666, 1740, 1691, 1596, 1575, 1485, 1446, 1392, 1304, 1265, 1148, 1078, 1020, 908, 829, 796, 752, 687, 640. UV (EtOH) λ_max_, (lgε): 246 (3.45), 301 (2.76), 331 (3.51), 357 (3.23) nm. ^1^H-NMR (CDCl_3_+CD_3_OD, 400 MHz, δ_H_): 4.93 (s, 2Н, СН_2_), 6.22 (d, *J* = 9.8 Hz, 1Н, Н-3), 6.82 (s, 1Н, Н-8), 7.13 (dd, *J* = 7.6, 7.2 Hz, 1Н, Н-4′′), 7.18 (1Н, d, *J* = 8.0 Hz, 1Н, Н-6′′), 7.48 (dd, *J* = 8.0, 7.2 Hz, 1Н, Н-5′′), 7.63 (d, *J* = 9.8 Hz, 1Н, Н-4), 7.78 (s, 1Н, Н-5), 7.88 (d, *J* = 7.6 Hz, 1Н, Н-3′′), 7.98 (s, 1Н, Н-5′). ^13^C-NMR (CDCl_3_+CD_3_OD, 100 MHz, δ_C_): 44.5, 104.3, 111.8, 113.4, 119.8, 121.3, 123.0, 124.1, 125.9, 129.5, 130.7, 132.1, 135.9, 142.3, 143.8, 158.5, 160.5, 163.8, 167.6, 169.2. Anal. calcd for C_20_H_14_N_4_O_6_: C, 59.12; H, 3.47; N, 13.79; found: C, 58.88; H, 3.77; N, 14.14.

*3-(4-((7-Hydroxy-2-oxo-2H-chromene-6-carboxamido)methyl)-1H-1,2,3-triazol-1-yl)benzoic acid* (**4b**). Yellowish powder, m.p. 186–187 °C (hexane). Yield 64%. IR (KBr, *ν*, cm^−1^): 3427, 3095, 3062, 2923, 2852, 2665, 1738, 1687, 1631, 1600, 1581, 1456, 1305, 1236, 1153, 1113, 1097, 1056, 1008, 931, 912, 893, 800, 795, 754, 698, 673. UV (EtOH) λ_max_, (lgε): 225 (3.66), 246 (3.53), 301 (3.23), 336 (3.22), 347 (3.18), 357 (3.23) nm. ^1^H-NMR (CDCl_3_, 400 MHz, δ_H_): 4.97 (s, 2Н, СН_2_), 6.30 (d, *J* = 9.8 Hz, 1Н, Н-3), 6.87 (s, 1Н, Н-8), 7.45 (d, *J* = 8.2 Hz, 1Н, Н-6′′), 7.47 (dd, *J* = 8.2, 7.6 Hz, 1Н, Н5′′), 7.62 (d, *J* = 9.8 Hz, 1Н, Н-4), 7.77 (s, 1Н, Н-5), 7.88 (d, *J* = 7.6 Hz, 1Н, Н-4′′), 8.01 (s, 2Н, Н-5′, 2′′), 11.19 (br.s, 3H, OH, NH). ^13^C-NMR (CDCl_3_, 100 MHz, δ_C_): 44.2, 104.9, 111.9, 114.2, 120.4, 121.7, 123.4, 124.3, 126.9, 127.1, 130.0, 130.7, 135.7, 140.8, 143.1, 158.9, 160.2, 161.0 (С-7), 164.2, 170.8. Anal. calcd for C_20_H_14_N_4_O_6_: C, 59.12; H, 3.47; N, 13.79; found: C, 59.55; H, 3.49; N, 13.43.

*4-(4-((7-Hydroxy-2-oxo-2H-chromene-6-carboxamido)methyl)-1H-1,2,3-triazol-1-yl)benzoic acid* (**4c**). Grayish powder, m.p. 203–204 °C (hexane). Yield 60%. IR (KBr, *ν*, cm^−1^): 3433, 3062, 2958, 2924, 2852, 2673, 1734, 1676, 1630, 1600, 1579, 1448, 1427, 1392, 1288, 1236, 1215, 1143, 1113, 1076, 946, 904, 858, 827, 796, 768, 732. UV (EtOH) λ_max_, (lgε): 225 (3.66), 246 (3.78), 270 (3.65), 293 (3.53), 324 (3.36), 351 (3.21) nm. ^1^H-NMR (CDCl_3_, 400 MHz, δ_H_): 4.95 (s, 2Н, СН_2_), 6.28 (d, *J* = 9.8 Hz, 1Н, Н-3), 6.87 (s, 1Н, Н-8), 7.10 (d, *J* = 8.4 Hz, 2Н, Н-2′′,6′′), 7.61 (d, *J* = 9.8 Hz, 1Н, Н-4), 7.98 (s, 1Н, Н-5), 8.02 (s, 1Н, Н-5′), 8.03 (br.s, 1Н, NН), 8.10 (d, *J* = 8.4 Hz, 2Н, Н-3′′,5′′), 11.19 (br.s, 2Н, ОН). ^13^C-NMR (CDCl_3_, 100 MHz, δ_C_): 42.5, 104.9, 111.9, 114.1, 118.9 (2C), 121.8, 125.3, 125.7, 127.6, 130.7, 132.1 (2С), 143.0, 145.7, 158.9, 160.1, 164.2, 169.5, 170.7. Anal. calcd for C_20_H_14_N_4_O_6_: C, 59.12; H, 3.47; N, 13.79; found: C, 59.16; H, 3.14; N, 13.42.

*3-Bromopeuruthenicin* (**11**). A solution of 1.0 g (4.5 mmol) of peuruthenicin (**2**) and 2.48 g (9 mmol) of dioxane dibromide in CH_2_Cl_2_ (10 mL) was stirred at rt for 16 h. Then the reaction mixture was filtered, and the filtrate was evaporated. 3-Bromopeuruthenicin (**11**, yield 86%, 1.14 g) was finally purified by crystallization from chloroform. White solid; m.p. 187–188 °C (ether). IR (KBr, *ν*, cm^−1^): 3480, 3350, 3116, 3050, 2956, 2853, 1720, 1675, 1621, 1616, 1598, 1440, 1311, 1293, 1231, 1163, 1124, 1089, 1030, 989, 964, 918, 845, 783, 750, 729, 719. UV (EtOH) λ_max_, (lgε): 244 (4.19), 314 (4.02), 334 (4.10), 400 (3.39) nm. ^1^H-NMR (600 MHz, CDCl_3_, δ_H_): 3.99 (s, 3H, OCH_3_), 6.85 (s, 1H, H-8), 7.97 (s, 2H, H-4,5), 11.21 (s, 1H, OH). ^13^C-NMR (150 MHz, CDCl_3_, δ_C_): 52.9, 104.8, 108.8, 110.5, 112.5, 129.9, 143.8, 156.4, 157.9, 164.3, 169.2. HRMS (ESI), *m*/*z* (*I*_rel_, %): 298(66), 269(15.9), 240(24), 220(18), 188(31), 160(16), 131(17), 103(19), 79(12); calcd for C_1__1_H_7_BrO_5_ [*M*]: 297.9471; found: 297.9468; Anal. Calcd. for C_11_H_7_BrO_5_: С, 44.18; Н, 2.36; Br, 26.72. Found: С, 44.08; Н, 2.16; Br, 26.70.

*3-Azidopeuruthenicin* (**12**). To a stirred solution of 25 mg (0.84 mmol) of 3-bromopeuruthenicin (**11**) in DMF at room temperature was added NaN_3_ (82 mg, 1.26 mmol). The reaction mixture was heated at 40 °C for 14h. After consumption of the bromocoumarin, the mixture was cooled to room temperature, quenched with eq. NaCl (10 mL), and the products and were extracted with CH_2_Cl_2_ (5 × 5 mL). The combined organic phases were dried over anhydrous MgSO_4_, after filtration, the solvent was removed under reduced pressure and was further dried in a freeze drier at oil pump vacuum. (P = 1 mm). The product (0.15 g, 68%) was obtained as a yellow solid, m.p. 161–162 °C. IR (KBr, *ν*, cm^−1^): 3455, 3390, 3357, 3299, 3004, 2971, 2923, 2858, 2121, 1727, 1621, 1467, 1434, 1375, 1309, 1290, 1263, 1193, 975, 869, 728. UV (EtOH) λ_max_, (lgε): 223(4.51), 243(4.4), 329(4.2), 373(3.97). ^1^H-NMR (CDCl_3_, 400 MHz, δ_H_): 3.99 (s, 3H, OCH_3_), 6.83 (s, 1H, H-8), 7.58 (s, 1Н, H-4), 8.00 (s, 1H, H-5), 11.19 (s, 1H, OH). ^13^C- NMR (CDCl_3_, 100 MHz, δ_C_): 52.7, 104.8, 109.9, 111.9, 114.0, 120.6, 142.9, 158.8, 159.9, 164.2, 169.4. Anal. Calcd for C_11_H_7_N_3_O_5_: С, 50.58; Н, 2.70; N, 16.09; found С, 51.04; Н, 3.12; N, 15.74.

#### 4.2.2. General Method for the N-Propargylation of Amino Acid Methyl Esters

Propargyl bromide (10.5 mmol, 80% in toluene solution) was added to a solution of the appropriate amino acid ether hydrochlorides (10 mmol) and potassium carbonate (21 mmol) in dry DMF at room temperature under argon. The mixture was stirred for 1–3 days (TLC control), the solvent was evaporated under reduced pressure and the residue partitioned between water (30 mL) and CH_2_Cl_2_ (30 mL). The organic phase was separated and the aqueous phase was extracted with CH_2_Cl_2_ (3 × 10 mL). The organic phases were combined, dried (MgSO_4_), filtered and the solvent evaporated under reduced pressure. The residue was purified by column chromatography to give secondary amines **13**, **14**, **15**.

*Methyl 9-(prop-2-ynylamino)nonanoate* (**13**)*. Yellowish oil.* 1H-NMR (CDCl_3_, 400 MHz, δ_H_): 1.22 (m, 4H, CH_2_-11′,12′), 1.39 (2H, CH_2_-13′), 1.52 (2H, CH_2_-10′), 2.12 (4H, CH_2_-9′,14′), 2.21 (m, 2H, CH_2_-15′), 2.58 (2H, CH_2_-8′), 3.32 (s, 2H, CH_2_-6′), 3.57 (s, 3H, OCH_3_).^13^C-NMR (CDCl_3_, 100 MHz, δ_C_): 25.2, 27.4, 29.3, 29.5, 30.0, 34.32, 38.40, 48.90 (CH_2_-12′,9′,5′), 51.62 (OCH_3_), 71.38, 73.02 (C≡CH). Anal. calcd for C_13_H_23_NO_2_: С, 69.29; Н, 10.29; N, 6.22; found С, 68.74; Н, 10.10; N, 5.94.

*Methyl 7-hydroxy-3-(4-((9-methoxy-9-oxononylamino)methyl)-1H-1,2,3-triazol-1-yl)-2-oxo-2H-chromene-6-carboxylate* (**8**). A solution of 3-azidopeuruthenicin (**12**, 50 mg, 0.19 mmol) in methylene chloride (10 mL) and a solution of sodium ascorbate (15 mol%, 0.255 mmol) and СuSO_4_ × 5H_2_O (5 mol%, 0.085 mmol) in water (10 mL) were mixed and the terminal alkyne **13** (63 mg, 0.28 mmol) was added with stirring. The reaction mixture was stirred for 3 h at room temperature and then for 1 h at 40 °С. The cooled mixture was diluted with water (10 mL) and the product was extracted with methylene chloride (4 × 10 mL). The combined extracts were dried over MgSO_4_, and the solvent was removed in vacuo. The residue was purified by column chromatography to give 73 mg (67%) of compound **8** as a yellow oil. IR (KBr, *ν*, cm^−1^): 3465, 3396, 3118, 3047, 2954, 2923, 2852, 1751, 1735, 1673, 1623, 1602, 1573, 1469, 1438, 1367, 1288, 1230, 1162, 1124, 1089, 1031, 962, 943, 865, 792. UV (EtOH) λ_max_, (lgε): 242 (4.49), 299 (4.17), 313 (4.26), 334 (4.34), 390 (3.39) nm. ^1^H-NMR (CDCl_3_, 400 MHz, δ_H_): 1.25 (m, 4H, CH_2_-11′,12′), 1.66 (2H, CH_2_-13′), 2.18 (2H, CH_2_-10′), 2.40 (4H, CH_2_-9′,14′), 2.47 (m, 2H, CH_2_-15′), 2.72 (2H, CH_2_-8′), 3.65 (s, 3H, OCH_3_), 3.99 (s, 3H, OCH_3_), 4.02 (s, 2H, CH_2_-6′), 6.87 (s, 1H, H-8), 7.97 (s, 1Н, H-4), 7.98 (s, 1H, H-5), 8.01 (s, 1H, H-5′), 11.25 (br.s, 2H, OH, NH). ^13^C-NMR (CDCl_3_, 100 MHz, δ_C_): 29.7, 30.8, 31.9, 35.5, 39.6, 40.1, 40.5, 45.1, 52.9, 61.9, 104.7, 108.7, 110.5, 112.5, 121.4, 124.9, 142.2, 143.8, 156.4, 157.8, 164.3, 169.2, 174.4. Anal. calcd. for C_24_H_30_N_4_O_7_: C, 59.25; H, 6.22; N, 11.52; found: C, 59.02; H, 6.12; N, 11.27.

#### 4.2.3. Preparation of 3-(Triazolyl)coumarins **9**, **10**

A stirred solution of 3-azidopeuruthenicin (**12**, 90 mg, 0.35 mmol) in MeCN (15 mL) was successively treated with alkyne **14** or **15** (1.2 equiv.), CuI (0.1 equiv.) and Et_3_N (1.1 equiv.) and the reaction mixture was stirred at room temperature until complete consumption of the starting material. Then, water (5 mL) was added and the products were extracted with CH_2_Cl_2_ (3 × 10 mL). The combined organic phases were dried (MgSO4), filtered and concentrated in vacuo. The residue was purified by column chromatography, eluting with chloroform, chloroform:ethanol 100:1.

*Methyl 7-hydroxy-3-(4-((1-methoxy-1-oxobutan-2-ylamino)methyl)-1H-1,2,3-triazol-1-yl)-2-oxo-2H-chromene-6-carboxylate* (**9**). Yield 56%. Yellow oil. IR (KBr, *ν*, cm^−1^): 3463, 3398, 3307, 3050, 2954, 2923, 2852, 1733, 1677, 1625, 1575, 1465, 1440, 1367, 1309, 1292, 1232, 1220, 1164, 1124, 1089, 1012, 968, 865, 794, 948. UV (EtOH) λ_max_, (lgε): 207 (4.69), 244 (4.29), 300 (3.95), 315 (4.02), 336 (4.1), 387 (3.41) nm. ^1^H-NMR (CDCl_3_+CD_3_OD, 400 MHz, δ_H_): 0.81 (t, 3H, *J* = 7.0 Нz, CH_3_-10′), 31.55 (m, 2H, CH_2_-9′), 3.27 (m, 1H, H-8′), 3.62 (s, 3H, OCH_3_), 3.98 (s, 3H, OCH_3_), 3.90 (m, 2H, H-6′), 6.76 (s, 1H, H-8), 7.90 (s, 1Н, H-4), 7.91 (s, 1H, H-5), 8.09 (s, 1H, H-5′). ^13^C-NMR (CDCl_3_, 100 MHz, δ_C_): 9.5, 25.8, 36.3, 51.6, 51.8, 60.7, 104.4, 108.2, 110.3, 112.3, 121.3, 124.8, 141.0, 144.1, 156.5, 157.5, 164.0, 168.9, 174.8. Anal. calcd. for C_19_H_20_N_4_O_7_: C, 54.81; H, 4.84; N, 13.46; found, %: C, 55.13; H, 5.02; N, 13.59.

*(S)-Methyl 7-hydroxy-3-(4-((1-methoxy-1-oxo-3-phenylpropan-2-ylamino)methyl)-1H-1,2,3-triazol-1-yl)-2-oxo-2H-chromene-6-carboxylate* (**10**). Yield 62% (by method for snthesis of compound 8–34%). Yellow oil. [α]_D_ +12.04 (*с* 1.00, CHCl_3_). IR (*ν*, cm^−1^): 3459, 3395, 3290, 3049, 2954, 2925, 2854, 1737, 1673, 1623, 1602, 1573, 1537, 1490, 1438, 1367, 1288, 1232, 1162, 1124, 1087, 962, 943, 865, 792. UV (EtOH) λ_max_, (lgε): 244 (4.22), 312 (3.94), 334 (4.02), 399 (3.26) nm. ^1^H-NMR (CDCl_3_, 400 MHz, δ_H_): 2.90-3.03 (m, 2H, H-9′), 3.36 (dd, *J* = 16.3, 1.8 Hz, 2H, H-6′), 3.65 s (3H, OCH_3_), 3.73 (dd, *J* = 6.4, 7.0 Hz, 1H, H-8′), 3.98 (s, 3H, OCH_3_), 6.84 s (1H, H8), 7.16-7.18 (m, 2H, *o*-Ph), 7.19-7.22 (m, 1H, *p*-Ph), 7.26-7.32 (m, 2H, *m*-Ph), 7.93 s (1Н, H-4), 7.95 (s, 1H, H-5), 8.08 (s, 1H, H-5′), 11.13 (br s, 2H, OH, NH). ^13^C-NMR (CDCl_3_, 100 MHz, δ_C_): 36.7, 39.3, 51.8, 52.8, 61.0, 104.9, 108.4, 110.3, 112.4, 120.5 (C-4), 125.1, 126.9, 129.1 (2C), 129.5 (2C), 132.5, 141.3, 143.8, 156.1, 157.7, 164.2, 169.2, 174.2. Anal. calcd. for C_24_H_22_N_4_O_7_: C, 60.25; H, 4.63; N, 11.71; found: C, 60.16; H, 4.96; N, 11.55.

#### 4.2.4. General Procedure for the Sonogashira Cross-coupling Reactions of 3-Bromopeuruthenicin **11** with Aryl Alkynes **17**, **19**–**24** or 2-methylbut-3-yn-2-ol **31**

To a solution of 3-bromopeurutenicin (**11**, 100 mg, 0.34 mmol) and terminal alkynes **17**, **19**–**24** or **31** (0.54 mmol) in benzene (5 mL) was added CuI (6 mg, 10 mol%), Pd(PPh_3_)_2_Cl_2_ (12 mg, 5 mol%), and Et_3_N (0.076 mL, 0.44 mmol; 1.3 (eq. to **11**) under argon. The reaction mixture was stirred at 80 °C for 12–14 h (TLC). The mixture was cooled and 5 mL of water was added. The separated water layer was extracted with СН_2_Cl_2_ (5 × 4 mL). The combined organic extracts was washed with water, dried over MgSO_4_, filtered, and concentrated under reduced pressure. The residue was subjected to column chromatography on silica gel. Eluting with chloroform and crystallization from ether afforded the corresponding 3-(*R*)-alkynylcoumarins **18, 25**–**30, 32**.

*Methyl 7-hydroxy-2-oxo-3-(phenylethynyl)-2H-chromene-6-carboxylate* (**18**). Yield 65% (70 mg). Yellow oil. IR (*ν*, cm^−1^): 3430, 3350, 3052, 2954, 2925, 2852, 2112 (С≡С), 1736, 1682, 1616, 1571, 1442, 1367, 1290, 1224, 1160, 1128, 1091, 966, 873, 792, 750. UV (EtOH) λ_max_, (lgε): 237 (4.14), 272 (3.86), 280 (3.88), 315 (3.78), 335 (3.83) nm. ^1^H-NMR (CDCl_3_, 400 MHz, δ_H_): 3.85 (s, 3H, OCH_3_), 6.79 (s, 1H, H-8), 7.48–7.56 (m, 5H, H-Ph), 7.92 s(1H, H-4), 7.93 (s, 1H, H5), 11.21 (1H, OH). ^13^C-NMR (CDCl_3_, 100 MHz, δ_C_): 52.1, 83.3, 95.2, 108.1, 112.0, 113.1, 114.7, 129.8 (2C), 130.0, 130.8, 132.8, 133.6 (2C), 144.4, 157.6, 158.4, 164.6, 169.7. Anal. Calcd for C_19_H_12_O_5_: 320.0745; found: 320.0743.

*Methyl 3-((4-fluorophenyl)ethynyl)-7-hydroxy-2-oxo-2H-chromene-6-carboxylate* (**25**). Yield 68% (78 mg), yellowish powder, m.p. 167–168 °C (ether). IR (KBr, *ν*, cm^−1^): 3425, 2933, 2836, 2145, 1733, 1673, 1608, 1517, 1463, 1450, 1373, 1346, 1257, 1228, 1124, 1012, 939, 873, 852, 840, 792, 753. UV (EtOH) λ_max_, (lgε): 218 (4.56), 225 (4.54), 245 (4.64), 305 (4.3), 329 (4.41) nm. ^1^H-NMR (CDCl_3_, 400 MHz, δ_H_): 3.91 (s, 3H, OCH_3_), 6.77 (s, 1H, H-8), 6.93 (dd, *J* = 8.2, 7.2 Нz, 2H, H-3′,5′), 7.40 (dd, *J* = 8.2, 3.4 Нz, 2H, H-2′,6′), 7.74 (s, 1H, H-4), 7.92 (s, 1H, H-5), 11.26 (1H, OH). ^13^C-NMR (CDCl_3_, 100 MHz, δ_C_): 53.3, 82.0, 95.7, 104.7, 110.7, 111.2, 112.1, 118.9 (2C), 128.6, 130.5, 131.8 (2C), 143.4, 158.0, 160.1(C-4′, J_C-F_ 259.1 Hz), 161.3, 164.4, 169.4. Anal. calcd for C_19_H_11_FO_5_: С, 67.46; Н, 3.28; F, 5.62; found: С, 68.08; Н, 3.50; F, 5.12.

*Methyl 7-hydroxy-2-oxo-3-(p-tolylethynyl)-2H-chromene-6-carboxylate* (**26**). Yield 61% (70 mg). brownish powder, m.p. 132–133 °C (ether). IR (KBr, *ν*, cm^−1^): 3430, 3299, 3259, 3049, 2956, 2852, 2121, 1753, 1735, 1697, 1677, 1594, 1516, 1436, 1367, 1321, 1294, 1234, 1203, 1162, 1087, 960, 848, 794, 748, 734. UV (EtOH) λ_max_, (lgε): 237 (4.41), 251 (4.21), 272 (4.13), 281 (4.16), 314 (4.04), 338 (4.11) nm. ^1^H-NMR (CDCl_3_, 400 MHz, δ_H_) 2.36 (s, 3H, CH_3_), 4.00 (s, 3H, OCH_3_), 6.88 (s, 1H, H-8), 7.15 (d, *J* = 8.2 Нz, 2H, H3′,5′), 7.45 (d, *J* = 8.2 Нz, 2H, H-2′,6′), 7.81 (s, 1H, H-4), 8.00 (s, 1H, H-5), 11.24 (1H, OH). ^13^C-NMR (CDCl_3_, 100 MHz, δ_C_): 21.7, 52.9, 82.4, 95.9, 104.8, 110.5, 111.2, 112.1, 119.00, 129.2 (2C), 130.5, 131.8 (2C), 139.3, 143.9, 158.0, 158.8, 164.4, 169.5. Anal. calcd for C_20_H_14_O_5_: С 71.85, Н 4.22, found: С 71.36, Н 3.94.

*Methyl 3-((4-ethylphenyl)ethynyl)-7-hydroxy-2-oxo-2H-chromene-6-carboxylate* (**27**). Yield 52% (61 mg) brownish powder, m.p. 130–131 °C (ether). IR (KBr, *ν*, cm^−1^): 3380, 3139, 3049, 2929, 2856, 2185, 1736, 1673, 1625, 1579, 1481, 1439, 1389, 1351, 1288, 1230, 1162, 1126, 1091, 1037, 962, 946, 865, 825, 750, 736. UV (EtOH) λ_max_, (lgε): 225 (4.1), 249 (3.67), 259 (3.59), 335 (4.18) nm. ^1^H-NMR (CDCl_3_, 400 MHz, δ_H_): 1.21 (t, *J* = 6.8 Нz, 3H, Et), 2.62 (q, *J* = 6.8 Нz, 2H, Et), 3.97 (s, 3H, OCH_3_), 6.85 (s, 1H, H-8), 7.13 (dd, *J* = 8.4 Нz, 2H, H-3′,5′), 7.40 (d, *J* = 8.4 Нz, 2H, H-2′,6′), 7.96 (s, 1H, H4), 7.97 (s, 1H, H5), 11.21 (1H, OH). ^13^C-NMR (CDCl_3_, 100 MHz, δ_C_): 17.3, 28.2, 53.5, 81.9, 97.3, 105.6, 112.1, 113.5, 114.8, 122.9, 129.2, 129.9 (2C), 132.6 (2C), 144.2, 144.5, 158.8, 159.5, 165.2, 170.2. Anal. calcd for C_21_H_16_O_5_: С, 72.41; Н, 4.63; found: С, 72.18; Н, 4.90.

*Methyl 7-hydroxy-3-((4-methoxyphenyl)ethynyl)-2-oxo-2H-chromene-6-carboxylate* (**28**). Yield 58% (70 mg) brownish powder, m.p. 134–135 °C (ether). IR (KBr, *ν*, cm^−1^): 3430, 3303, 3216, 3049, 3014, 2956, 2129, 1732, 1677, 1594, 1516, 1491, 1436, 1367, 1321, 1294, 1267, 1234, 1203, 1163, 1124, 1088, 960, 865, 849, 794, 748, 734. UV (EtOH) λ_max_, (lgε): 237 (4.18), 265 (3.78), 277 (3.91), 280 (3.92), 324 (3.84), 335 (3.88) nm. ^1^H-NMR (CDCl_3_, 400 MHz, δ_H_): 3.91 (s, 3H, OCH_3_), 3.96 (s, 3H, OCH_3_), 6.86 (s, 1H, H-8), 7.13 (d, *J* = 8.6 Нz, 2H, H-3′,5′), 7.41 (d, *J* = 8.6 Нz, 2H, H-2′,6′), 7.97 (s, 1H, H-4), 7.98 (s, 1H, H-5), 11.18 (1H, OH). ^13^C-NMR (CDCl_3_, 100 MHz, δ_C_): 53.2, 53.6, 81.6, 95.4, 105.1, 111.1, 111.7, 112.8, 118.7, 119.4 (2C), 129.5, 133.3 (2C), 144.1, 157.9, 158.7, 161.6, 164.7, 169.8. Anal. calcd for C_20_H_14_O_6_: С, 68.57; Н, 4.03; found: С, 68.88; Н, 3.90.

*Methyl 3-((4-acetylamino-3-(methoxycarbonyl)phenyl)ethynyl)-7-hydroxy-2-oxo-2H-chromene-6-carboxylate* (**29**). Yield 45% (66 mg) yellowish powder, m.p. 154–155 °C (ethanol). IR (KBr, *ν*, cm^−1^): 3430, 3299, 3259, 3216, 3049, 2956, 2925, 2852, 2208, 1753, 1735, 1697, 1677, 1620, 1594, 1515, 1491, 1437, 1367, 1321, 1294, 1234, 1203, 1162, 1124, 1088, 960, 920, 866, 849, 795, 748, 735. UV (EtOH) λ_max_, (lgε): 237 (4.28), 272 (4.00), 280 (4.01), 316 (3.92), 335 (3.96) nm. ^1^H-NMR (CDCl_3_, 400 MHz, δ_H_): 2.19 (s, 3H, CH_3_-CO), 3.88 (s, 3H, OCH_3_), 3.95 (s, 3H, OCH_3_), 6.82 (s, 1H, H-8), 7.55 (dd, *J* = 8.8, 2.0 Нz, 1H, H-6′), 7.94 (s, 1H, H-4), 7.95 (s, 1H, H-5), 8.09 (d, *J* = 2.0 Нz, 1H, H-2′), 8.58 (d, *J* = 8.8 Нz, 1H, H-5′), 11.16 (br.s, 2H, NH, OH). ^13^C-NMR (CDCl_3_+CD_3_OD, 100 MHz, δ_C_): 25.2, 52.5, 52.8, 71.3, 82.2, 104.6, 108.5, 112.4, 116.2, 120.0, 130.0, 130.5, 133.7, 134.6, 137.7, 141.2, 143.9, 156.5, 157.7, 166.4, 167.9, 169.1, 169.3. Anal. calcd for C_23_H_17_NO_8_: С, 63.45; Н, 3.94; N, 3.22; found: С, 63.18; Н, 3.92; N, 3.12.

*Methyl 7-hydroxy-2-oxo-3-(pyridin-2-ylethynyl)-2H-chromene-6-carboxylate* (**30**). Yield 55% (60 mg) brown oil. IR (*ν*, cm^−1^): 3444, 2971, 2929, 2856, 2133, 1735, 1674, 1626, 1579, 1481, 1438, 1388, 1352, 1300, 1230, 1163, 1126, 1092, 1038, 962, 947, 865, 825, 750, 736, 723. UV (EtOH) λ_max_, (lgε): 227 (4.25), 238 (4.15), 294 (3.93), 306 (3.92), 351 (4.13) nm. ^1^H-NMR (CDCl_3_, 400 MHz, δ_H_): 3.76 (s, 3H, OCH_3_), 6.72 (s, 1H, H-8), 7.18 (ddd, *J* = 7.4, 5.2, 1.4 Нz, 1H, H-4′), 7.31 (dd, *J* = 6.6, 1.4 Нz, 1H, H-6′), 7.54 (ddd, *J* = 7.4, 6.6, 1.8 Нz, 1H, H-5′), 7.69 (s, 1H, H-4), 7.77 (s, 1H, H-5), 8.49 (ddd, *J* = 7.2, 1.8 Нz, 1H, H-3′), 11.20 (1H, OH). ^13^C-NMR (CDCl_3_, 100 MHz, δ_C_): 52.9, 82.6, 94.8, 105.0, 112.8, 115.1, 116.6, 123.5, 129.4, 130.4, 134.1, 138.1, 141.6, 144.4, 156.9, 158.1, 164.4, 169.7. Anal. calcd for C_18_H_11_NO_5_: С, 67.29; Н, 3.45; N, 4.36; found: С, 67.08; Н, 3.38; N, 4.22.

*Methyl 7-hydroxy-3-(3-hydroxy-3-methylbut-1-ynyl)-2-oxo-2H-chromene-6-carboxylate* (**32**)**.** Yield 66% (68 mg), m.p.114–115 °C (ethanol). IR (KBr, *ν*, cm^−1^): 3282, 3221, 2981, 2932, 2868, 2250, 1741, 1679, 1612, 1600, 1584, 1462, 1446, 1363, 1296, 1269, 1210, 1170, 1120, 1096, 1058, 998, 964, 954, 900, 889, 842, 792, 735, 723. UV (EtOH) λ_max_, (lgε): 223 (4.04), 246 (3.84), 338 (3.67), 365 (3.57) nm. ^1^H-NMR (CDCl_3_, 400 MHz, δ_H_): 1.34 (s, 6H, 2 × CH_3_), 3.95 (s, 3H, OCH_3_), 6.78 (s, 1H, H-8), 7.76 (br. s, 1H, H-4), 7.93 (s, 1H, H-5). ^3^C-NMR (CDCl_3_, 100 MHz, δ_c_): 30.9, 31.0, 52.7, 65.3, 66.2, 84.1, 104.8, 109.9, 110.5, 114.0, 130.6, 143.1, 157.8, 160.0, 164.1, 169.4. Anal. calcd for C_16_H_14_O_6_: С, 63.57; Н, 4.67; found: С, 63.22; Н, 4.44. HRMS, *m/z* calcd.: 302.0785; found 302.0783 [*M*].

*Methyl 7-hydroxy-2-oxo-3-((trimethylsilyl)ethynyl)-2H-chromene-6-carboxylate* (**34**). A sealed 10 mL glass tube containing 3-bromopeuruthenicin (**11**, 100 mg (0.34 mmol), toluene (3 mL), trimethylsilylacetylene (**33**, 0.1 mL, 0.68 mmol), CuI (3 mg, 5 mol %), Pd(PPh_3_)_2_Cl_2_ (23 mg, 10 mol. %), and Et_3_N (0.061 mL, 1.3 eq.) was placed in the cavity of a microwave reactor and irradiated for 2 h at 100 °C and power 50 W in an Anton Paar Microwave 50 reactor. After cooling to 25 °C, the tube was removed from the reactor. Then the mixture was cooled and 5 mL of water was added. The separated water layer was extracted with СН_2_Cl_2_ (5 × 4 mL). The combined organic extracts was washed with water, dried over MgSO_4_, filtered, and concentrated under reduced pressure. The residue was crystallized from ether, yield 64% (68 mg), m.p. 104–107 °C. IR (KBr, *ν*, cm^−1^): 3437, 2924, 2852, 2223, 1735, 1672, 1314, 1597, 1508, 1425, 1388, 1328, 1236, 1222, 1159, 1107, 908, 839, 752, 694, 680. UV (EtOH) λ_max_, (lgε): 325 (3.97), 304 (3.85), 241 (4.01) nm. ^1^H-NMR (CDCl_3_, 400 MHz, δ_H_) 0.20 (s, 9H, SiMe_3_), 3.93 (s, 3H, OCH_3_), 6.76 (s, 1H, H-8), 7.74 (s, 1H, H-4), 7.91 (s, 1H, H-5), 10.97 (s, 1H, OH). ^13^C-NMR (CDCl_3_, 100 MHz, δ_C_): 8.6, 52.7, 73.8, 81.5, 101.6, 110.2, 110.4, 111.7, 128.4, 145.1, 158.0, 158.3, 164.45, 169.2. Anal. calcd for C_16_H_16_O_5_Si: С, 60.74; Н, 5.10; Si, 8.88; found: С, 60.55; Н, 5.18; Si, 8.54.

*Methyl 3-ethynyl-7-hydroxy-2-oxo-2H-chromene-6-carboxylate* (**16**). To a solution of compound **34** (60 mg, 0.00019 mol) in methanol (3 mL) were added CsF (14 mg, 0.00095 mol) and TEBA (5 mg). The mixture was stirred at rt for 10 h in an argon flow (TLC). Then 10 mL of water was added and the mixture was extracted with methylene chloride (5 × 4 mL). The combined extract was washed with water, dried over MgSO_4_, filtered, and concentrated under reduced pressure. The residue was subjected to column chromatography on silica gel. Eluting with chloroform and crystallization from ether gave compound **16**, yield 85%, m.p. 84- 88 ^o^C. IR (KBr, *ν*, cm^−1^): 3463, 2956, 2923, 2854, 2102, 1741, 1677, 1621, 1581, 1494, 1444, 1416, 1367, 1295, 1234, 1186, 1157, 1120, 1099, 1024, 952, 914, 856, 721. UV (EtOH) λ_max_, (lgε): 225 (4.1), 246 (3.96), 321 (3.84), 344 (3.92). ^1^H-NMR (CDCl_3_, 400 MHz) 2.16 (s, 1H, CH), 3.99 (s, 3H, OCH_3_), 6.86 (s, 1H, H-8), 7.83 (s, 1H, H-4), 8.00 (s, 1H, H-5), 11.29 (s, 1H, OH). ^13^C- NMR (CDCl_3_, 100 MHz, δ_C_): 52.3, 75.4, 82.2, 105.1, 112.6, 114.0, 114.8, 128.5, 143.3, 159.4, 159.7, 164.7, 168.9. Anal. calcd for C_13_H_8_O_5_ С 63.94, Н 3.30 found: С 63.65, Н 3.84.

*Methyl 7-hydroxy-3-[1-(2-isopropyl-3,7-dioxo-2,3-dihydro-7H-furo[3,2-g]chromen-2-yl)-1H-1,2,3-triazol-4-yl]-2-oxo-2H-chromene-6-carboxylate* (**36**). A solution of 2-azidooreoselone (**35**, 100 mg, 0.35 mmol) and 3-ethynylpeurutenicin (**16**, 85 mg, 0.35 mmol) in CH_2_Cl_2_ (10 mL) was mixed with a solution of sodium ascorbate (10 mg, 15 mol%) and СuSO_4_·5H_2_O (4 mg, 5 mol%) in Н_2_О (5 mL). The reaction mixture was stirred for 3 h at rt and additional for 1 h at 40 °С. After the completion of the reaction, the mixture was diluted with Н_2_О (10 mL), the product was extracted with CH_2_Cl_2_ (4 × 10 mL), the combined extracts were dried over anhydrous MgSO_4_, filtered, and the solvent was removed at reduced pressure. Yield 70% (0.130 g), yellowish solid. IR (KBr, *ν*, cm^−1^): 3469, 3049, 2983, 2956, 2883, 2850, 1739, 1675, 1629, 1556, 1467, 1438, 1390, 1353, 1284, 1132, 1162, 1128, 1093, 1025, 964, 944, 869, 827, 736. UV (EtOH) λ_max_, (lgε): 217 (4.57), 249 (4.59), 294 (4.26), 307 (4.27), 337 (4.27), 381 (3.76), 393 (3.36) nm. ^1^H-NMR (CDCl_3_, 400 MHz, δ_H_): 0.89 (d, *J* = 7.0 Нz, 3H, CH_3_), 1.11 (d, *J* = 7.0 Нz, 3H, CH_3_), 3.30 (m, 1H, CH(Me)_2_), 3.99 (s, 3H, OCH_3_), 6.41 (d, *J* = 9.6 Нz, 1H, H-6), 6.88 (s, 1H, H-8′), 7.24 (s, 1H, H-9), 7.71 (d, *J* = 9.6 Нz, 1H, H-5), 7.87 (s, 1H, H-4), 7.97 (s, 1H, H-4′), 7.98 (s, 1H, H-5′), 8.13 (s, 1H, H-1′′), 11.29 (s, 1H, OH). ^13^C-NMR (CDCl_3_, 100 MHz, δ_C_): 16.2, 16.3, 30.1, 53.3, 96.5, 101.4, 105.2, 110.7, 111.8, 112.9, 115.7, 117.3, 117.5, 125.7, 128.0, 130.3, 142.9, 143.5, 144.3, 158.5, 159.3, 162.1, 164.7, 165.8, 169.6, 172.4, 194.7. Anal. calcd for C_27_H_19_N_3_O_9_: С, 61.25; Н, 3.62; N, 7.94; found: С, 61.01; Н, 3.54; N, 7.62.

#### 4.2.5. General Method for the Preparation of Monoalkynes **38a**–**c**

A solution of 2-azidooreoselone (**35**, 1.7 mmol) in methylene chloride (10 mL) and a solution of sodium ascorbate (15 mol%, 0.255 mmol) and СuSO4∙5H2O (5 mol%, 0.085 mmol) in water (10 mL) were mixed and the appropriate terminal alkyne **39**–**41** (0.85 mmol) was added with stirring. The reaction mixture was stirred for 5 h at 40 °С. The cooled mixture was diluted with water (10 mL) and the product was extracted with methylene chloride (4 × 10 mL). The combined extracts were dried over MgSO_4_, filtered, the solvent was removed to give the desired products **38a**–**с**.

*2-Isopropyl-2-(4-(pent-4-ynyl)-1H-1,2,3-triazol-1-yl)-2H-furo[3,2-g]chromene-3,7-dione* (**38a**). Yield 66% (0.420 g), brownish oil. IR (*ν*, cm^−1^): 3120, 2981, 2932, 2867, 2140, 1741, 1678, 1600, 1584, 1461, 1446, 1381, 1363, 1295, 1269, 1210, 1169, 1120, 1058, 964, 953, 909, 889, 792, 749. UV (EtOH) λ_max_, (lgε): 231 (3.91), 282 (3.58), 320 (3.52), 348 (3.43) nm. ^1^H-NMR (CDCl_3_, 400 MHz, δ_H_): 0.89 (d, *J* = 7.0 Hz, 3Н, СН_3_), 0.95 (d, *J* = 7.0 Hz, 3Н, СН_3_), 1.40 (m, 2H, CH_2_-7′), 1.85 (s, 1H, H-10′), 2.06 (m, 2H, CH_2_-6′), 2.55 (m, 2H, CH_2_-8′), 3.06 (m, 1H, CH(CH_3_)_2_), 6.28 (d, *J* = 9.6 Нz, 1H, H-6), 6.99 (s, 1H, H-9), 7.39 (s, 1H, H-4), 7.67 (d, *J* = 9.6 Нz, 1H, H-5), 7.82 (s, 1H, H-5′). ^13^C-NMR (CDCl_3_, 100 MHz, δ_C_): 15.4, 15.8, 19.5, 28.1, 28.2, 33.7, 69.2, 82.3, 98.4, 100.4, 115.4, 115.8, 117.0, 126.7, 127.5, 137.4, 145.9, 161.2, 164.0, 173.6, 191.5. Anal. calcd for C_21_H_19_N_3_O_4_: С, 66.83; Н, 5.07; N, 11.13; found: С, 67.01; Н, 5.12; N, 10.88.

*2-(4-(Hex-5-ynyl)-1H-1,2,3-triazol-1-yl)-2-isopropyl-2H-furo[3,2-g]chromene-3,7-dione* (**38b**). Yield 62% (0.415 g). Yellowish powder, m.p. 111–112 °C (ether). IR (KBr, *ν*, cm^−1^): 2971, 2828, 2856, 2115, 1735, 1673, 1625, 1579, 1481, 1438, 1388, 1351, 1230, 1126, 1091, 1037, 997, 962, 946, 865, 825, 750. UV (EtOH) λ_max_, (lgε): 221 (4.12), 255 (4.47), 292 (3.98), 308 (3.89), 351 (4.03) nm. ^1^H-NMR (CDCl_3_, 400 MHz, δ_H_): 0.89 (d, *J* = 7.0 Hz, 3Н, СН_3_), 1.03 (m, 2H, CH_2_-8′), 1.12 (d, *J* = 7.0 Hz, 3Н, СН_3_), 1.24 (m, 2H, CH_2_-7′), 1.92 (s, 1H, H-11′), 2.14 (m, 2H, CH_2_-6′), 2.30 (m, 1H, CH(СH_3_)_2_), 3.40 (m, 2H, CH_2_-9′ ), 6.35 (d, *J* = 9.6 Нz, 1H, H-6), 7.02 (s, 1H, H-9), 7.43 (s, 1H, H-4), 7.69 (d, *J* = 9.6 Нz, 1H, H-5), 7.84 (s, 1H, H-5′). ^13^C- NMR (CDCl_3_, 100 MHz, δ_C_): 14.9, 15.3, 17.8, 27.6, 27.7, 27.73, 33.2, 68.7, 81.8, 97.9, 100.6, 114.9, 115.3, 116.1, 125.4, 126.9, 136.8, 144.7, 162.4, 165.6, 173.1, 191.0. Anal. calcd for C_22_H_21_N_3_O_4_: С, 67.51; Н, 5.41; N, 10.74; found: С, 67.17; Н, 5.21; N, 10.72.

*2-Isopropyl-2-(4-(oct-7-ynyl)-1H-1,2,3-triazol-1-yl)-2H-furo[3,2-g]chromene-3,7-dione* (**38c**). Yield 74% (0.527 g). Yellowish powder, m.p. 121–122 °C (ether). IR (KBr, *ν*, cm^−1^): 3143, 3084, 2976, 2935, 2858, 2113, 1735, 1627, 1581, 1481, 1419, 1390, 1352, 1286, 1226, 1128, 1093, 1039, 997, 947, 866, 825, 739. UV (EtOH) λ_max_, (lgε): 221 (4.16), 255 (4.5), 294 (4.00), 307 (3.94), 339 (4.05), 351 (4.06) nm. ^1^H-NMR (CDCl_3_, 400 MHz, δ_H_): 0.94 (d, *J* = 7.0 Hz, 3Н, СН_3_), 0.97 (d, *J* = 7.0 Hz, 3Н, СН_3_), 1.27 (m, 2H, CH_2_-8′), 1.34 (m, 2H, CH_2_-9′),1.46 (m, 2H, CH_2_-10′), 1.55 (m, 2H, (m, 2H, CH_2_-7′), 1.89 (s, 1H, H-13′), 2.12 (m, 2H, CH_2_-6′), 2.60 (m, 2H, CH_2_-11′), 3.09 (m, 1H, CH(CH_3_)_2_), 6.32 (d, *J* = 9.6 Нz, 1H, H-6), 7.04 (s, 1H, H-9), 7.44 (s, 1H, H-4), 7.72 (d, *J* = 9.6 Нz, 1H, H-5), 7.87 (s, 1H, H-5′). ^13^C-NMR (CDCl_3_, 100 MHz, δ_C_): 15.1, 15.5, 18.1, 27.9, 28.0, 28.4, 28.67, 28.73, 33.5, 68.1, 84.4, 98.1, 100.8, 115.1, 115.5, 116.3, 125.7, 128.4, 137.7, 143.1, 158.5, 161.4, 171.4, 191.6. Anal. calcd for C_24_H_25_N_3_O_4_: С, 68.72; Н, 6.01; N, 10.02; [M] 419; found: С, 69.08; Н, 6.06; N, 10.27; М = 419.

#### 4.2.6. Sonogashira Cross-Coupling Reactions of Bromide **11** with Alkynes **38a**–**c**

To a solution of 3-bromopeurutenicin (**11**, 100 mg, 0.34 mmol) and a terminal alkyne **38a**–**c** (0.54 mmol) in benzene (5 mL) was added CuI (6 mg, 10 mol%), Pd(PPh_3_)_2_Cl_2_ (12 mg, 5 mol%), and Et_3_N (0.076 mL, 0.44 mmol; 1.3 equiv) under argon. The reaction mixture was stirred at 80 °C for 14 h (TLC). The mixture was cooled and 5 mL of water was added. The separated water layer was extracted with СН_2_Cl_2_ (5 × 4 mL). The combined organic extracts was washed with water, dried over MgSO_4_, filtered, and concentrated under reduced pressure. The residue was subjected to column chromatography on silica gel (eluent—chloroform).

*Methyl 7-hydroxy-3-(5-(1-(2-isopropyl-3,7-dioxo-3,7-dihydro-2H-furo[3,2-g]chromen-2-yl)-1H-1,2,3-triazol-4-yl)-pent-1-ynyl)-2-oxo-2H-chromene-6-carboxylate* (**37a**). Yield 52% (0.105 g). Brownish oil. IR (ν, cm^−1^): 3350, 3200, 2981, 2932, 2867, 2254, 1741, 1678, 1620, 1600, 1580, 1461, 1446, 1381, 1363, 1295, 1269, 1210, 1169, 1120, 1096, 1058, 964, 953, 889, 792, 735, 723. UV (EtOH) λ_max_, (lgε): 234 (4.41), 256 (4.44), 273 (4.31), 310 (4.09), 339 (4.14), 354 (4.08) nm. ^1^H-NMR (CDCl_3_, 400 MHz, δ_H_): 0.99 (d, *J* = 7.0 Hz, 3Н, СН_3_), 1.02 (d, *J* = 7.0 Hz, 3Н, СН_3_), 1.60 (m, 2H, CH_2_-7′), 2.18 (m, 2H, CH_2_-6′), 2.65 (m, 2H, CH_2_-8′), 3.13 (m, 1H, CH(CH_3_)_2_), 3.97 (s, 3H, OCH_3_), 6.37 (d, *J* = 9.8 Hz, 1H, H-6), 6.84 (s, 1H, H-8′′), 7.08 (s, 1H, H-9), 7.48 (d, 1H, *J* = 9.8 Hz, H-5), 7.76 (s, 1H, H-4), 7.71, 7.74 (both s, 2H, H-4′′,5′′), 7.97 (br.s, 2H, H-5′, OH). ^13^C-NMR (CDCl_3_, 100 MHz, δ_C_): 15.0, 15.4, 20.0, 23.5, 32.4, 34.7, 55.7, 70.0, 85.7, 98.1, 99.9, 104.3, 107.2, 110.2, 112.3, 115.1, 116.0, 116.3, 125.5, 128.2, 130.7, 139.7, 142.9, 143.8, 156.1, 157.5, 158.5, 161.4, 162.0, 169.0, 171.4, 192.5. Anal. calcd for C_32_H_25_N_3_O_9_: С, 64.54; Н, 4.23; N, 7.06; [M] 598; found: С, 64.58, Н, 4.36, N, 7.07; М = 595.

*Methyl 7-hydroxy-3-(6-(1-(2-isopropyl-3,7-dioxo-3,7-dihydro-2H-furo[3,2-g]chromen-2-yl)-1H-1,2,3-triazol-1-yl)-hex-1-ynyl)-2-oxo-2H-chromene-6-carboxylate* (**37b**). Yield 57% (0.118 g). Yellowish powder, m.p. 126–127 °C (ether). IR (KBr, *ν*, cm^−1^): 3388, 3139, 3049, 2929, 2856, 2235, 1735, 1673, 1626, 1579, 1481, 1439, 1388, 1352, 1288, 1230, 1163, 1126, 1091, 1037, 997, 962, 946, 865, 825, 750, 737. UV (EtOH) λ_max_, (lgε): 234 (4.39), 256 (4.43), 273 (3.81), 309 (4.09), 339 (4.13), 352 (3.62) nm. ^1^H-NMR (CDCl_3_, 400 MHz, δ_H_): 0.98 (d, *J* = 7.0 Hz, 3Н, СН_3_), 1.04 (d, *J* = 7.0 Hz, 3Н, СН_3_), 1.44–1.53 (m, 4H, CH_2_-7′,8′), 2.14 (m, 2H, CH_2_-6′), 2.64 (m, 2H, CH_2_-9′), 3.13 (m, 1H, CH(CH_3_)_2_), 3.95 (s, 3H, OCH_3_), 6.35 (d, *J* = 9.8 Hz, 1H, H-6), 6.92 (s, 1H, H-8′′), 7.07 (s, 1H, H-9), 7.64 (d, *J* = 9.8 Hz, 1H, H-5), 7.71, 7.74 (both s, 2H, H-4′′,5′′), 7.88 (s, 1H, H-4), 7.97 (s, 1H, H5′). ^13^C-NMR (CDCl_3_, 100 MHz, δ_C_): 15.6, 16.0, 19.4, 32.5, 33.0, 36.7, 25.7, 54.2, 70.6, 88.5, 98.7, 100.6, 104.9, 108.8, 110.8, 112.8, 115.7, 116.6, 116.9, 126.1, 128.8, 132.3, 141.1, 143.5, 144.4, 156.6, 158.1, 159.1, 162.0, 162.6, 169.5, 172.0, 192.4. Anal. calcd for C_33_H_27_N_3_O_9_: С, 65.02; Н, 4.46; N, 6.89; [M] 609; found: С, 65.08; Н, 4.06; N, 6.37; M = 605.

*Methyl 7-hydroxy-3-(8-(1-(2-isopropyl-3,7-dioxo-3,7-dihydro-2H-furo[3,2-g]chromen-2-yl)-1H-1,2,3-triazol-4-yl)-oct-1-ynyl)-2-oxo-2H-chromene-6-carboxylate* (**37c**). Brownish powder, m.p. 152–153 °C (ether). Yield 55% (0.119 g). IR (KBr, *ν*, cm^−1^): 3440, 3141, 3049, 2971, 2929, 2856, 2213, 1735, 1673, 1625, 1579, 1481, 1388, 1351, 1288, 1230, 1126, 1091, 1037, 962, 946, 865, 825, 750. UV (EtOH) λ_max_, (lgε): 234 (4.63), 251 (4.67), 256 (4.67), 309 (4.33), 339 (4.37), 350 (4.34) nm. ^1^H-NMR (CDCl_3_, 400 MHz, δ_H_): 0.91 (d, *J* = 7.0 Hz, 3Н, СН_3_), 0.94 (d, *J* = 7.0 Hz, 3Н, СН_3_), 1.23, 1.29 (both m, 4H, CH_2_-8′,9′), 1.42 (m, 2H, CH_2_-10′), 1.51 (m, 2H, CH_2_-7′), 2.07 (m, 2H, CH_2_-6′), 2.56 (m, 2H, CH_2_-11′), 3.07 (m, 1H, CH(CH_3_)_2_), 3.92 (s, 3H, OCH_3_), 6.31 (d, *J* = 9.8 Hz, 1H, H-6), 6.78 (s, 1H, H-8′′), 7.03 (s, 1H, H-9), 7.46 (s, 1H, H-4), 7.68 (d, *J* = 9.8 Hz, 1H, H-5), 7.83 (s, 1H, H-5′). 7.89, 7.90 (both s, 2H, H-4′′,5′′). ^13^C-NMR (CDCl_3_, 100 MHz, δ_C_): 15.2, 15.6, 19.0, 25.2, 29.9, 30.7, 31.2, 33.6, 36.3, 52.7, 70.2, 87.4, 98.2, 101.0, 104.5, 108.4, 110.4, 112.4, 115.2, 115.6, 116.5, 125.7, 128.2, 131.8, 141.3, 143.0, 144.0, 156.2, 157.7, 158.9, 161.6, 162.2, 169.2, 171.5, 191.8. Anal. calcd for C_35_H_31_N_3_O_9_: С, 65.93; Н, 4.90; N 6.59; [M] 637; found: С, 65.38; Н, 4.66; N, 6.87; М = 635.

### 4.3. Antibacterial Activity Assay

**Method 1**. Compounds **4a**–**c**, **11**, **26**, **29**, **30**, **37a**–**c**, **42a**–**c** and parent compounds **1**, **2, 3** were tested for their in vitro antibacterial activity against *Staphylococcus aureus* 209р ATCC 6538-P and strains from the collection of Department of Microbiology, Immunology and Virology, Novosibirsk State Medical University: *Staphylococcus aureus* С-18 - clinical isolated strain, *Staphylococcus aureus* “Viotko” and *Actinomyces viscosus* U-18. Cultivation of bacterial cultures was carried out on agar and broth media Muller-Hinton in aerobic conditions at a temperature of 37 °C. Cultivation time for *Staphylococcus aureus* was 1–2 days, for *Actinomyces viscosus* U – 2–3 days. Analysis of antibacterial activity was performed using the method of serial macrodilutions in a liquid medium in the total volume of 1.0 mL [24,38]. All compounds were primarily dissolved in 0.05 mL of 96% ethyl alcohol and brought to the desired concentration with 0.9% sodium chloride solution (0.2 mL) and nutrient broth. To assess antibacterial activity, a number of double dilutions of the studied substances starting from 1000 µg/mL were used. Doses of substances above 1000 µg/mL were not considered because of their low solubility. The introduced dose of daily cultures of bacteria was determined using the standard of turbidity according to McFarland and was controlled by seeding on a dense nutrient medium. It gave (2.75 ± 0.85) 10^3^ colony-forming units (CFU). For the minimum inhibitory concentration (MIC), the smallest dose of the substance was taken, completely suppressing the growth of bacteria. The absence of signs of growth in the liquid medium was controlled by seeding to the surface of a agar medium with subsequent incubation under standard conditions. As a negative control, the test culture was introduced into 1 mL of broth and cultivated under the same conditions, followed by sowing on an agar nutrient medium and taking into account the growth of bacteria. Based on the results of triplicate repeated experiments, the mean value of the MIC and the standard error of measurement (M ± SEM) were calculated. The primary data of antibacterial activity are given in Appendix A.

**Method 2**. Compounds **36**, **37а**–**с**, **1** and **2** were tested for their in vitro antibacterial activity against *Bacillus subtilis* and *Escherichia coli* (JM109). This strain is most similar to wild type of *E. coli* in our bacterial collection. The known tumorogenic compounds 4-nitroquinolin-1-oxide (NQO) was used as the reference compound [39]. This procedure was maintained according to the standard broth microdilution method as recommended in guidelines of Clinical and Laboratory Standards Institute [24,40,41] and the minimum inhibitory concentration (MIC) of compounds was tested. In short, testing was performed in U-bottomed 96-well sterile plastic microdilution trays (Falcon 3077, Becton Dickinson and Co., Cockeysville, MD, USA) in cation (Ca^2+^ and Mg^2+^) adjusted Mueller-Hinton broth medium (Becton Dickinson and Co., Cockeysville, MD, USA). The concentration range of test compounds was started from 1000 µg/mL by using serial two fold dilution. Standardized initial inoculum was prepared by the direct colony suspension method to the final inoculum to 5 × 10^5^ CFU/mL, as described (CLSI M7-A7). After inoculation of previously prepared microdilution trays with tested compounds, trays were incubated at 35 ± 2 °C overnight (18–20 h) in an ambient air incubator. *E. coli* JM109 served as quality control of MIC determination procedure, as well. The MIC was determined as the lowest concentration of tested compound that completely inhibits growth of the organism in the microdilution wells as detected by the unaided eye and comparing the amount of growth in the wells containing the tested agent with the amount of growth in the growth-control wells (no antimicrobial agent). All testing were done in triplicate.

### 4.4. Molecular Docking

Molecular modeling was carried out in the Schrodinger Maestro visualization environment using applications from the Schrodinger Small Molecule Drug Discovery Suite 2016-1 (Python, NY, USA) [42]. Three-dimensional structures of the derivatives were obtained empirically in the *LigPrep* application using the OPLS3 force field [43]. All possible tautomeric forms of compounds, as well as various states of polar protons of molecules in the pH range of 7.0 ± 2.0 were taken into account. The search area for docking was selected automatically, based on the size and physico-chemical properties of FAD. The extra precision (XP) algorithm of docking was applied. Docking was performed in comparison with FAD. The three-dimensional structures of FAD was obtained in the PubChem database and prepared in the LigPrep application. Non-covalent interactions of compounds in the binding site were visualized using Biovia Discovery Studio Visualizer (Biovia, San Diego, CA, USA) [44].

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
