# Peer review of "Design, Synthesis and Antibacterial Activity of Coumarin-1,2,3-triazole Hybrids Obtained from Natural Furocoumarin Peucedanin"

_molecules, 2019, doi:10.3390/molecules24112126_

Round 1
Reviewer 1 Report
The chemistry work was performed well and the authors need to fix these points especially the antibacterial assay which is considered a major revision:
Line 51, authors mentioned phthalimide, although there is no phthalimide in figure1.
Figure 1, authors need to show what is R, R1 and R2
In Scheme 1, compound 1 was not used, why its structure there?
Line 81, azide and not aside
Line 93, coumarin 6 and not 5.
All bacterial names should be italic and they need to include the ATCC numbers
Authors didn't include standard controls
What concentrations authors used in the antibacterial assay?
Where is the negative control?
Why did authors use E.coli JM109?
Authors showed the concentrations in uM and the MIC in ug/mL..how does this work?
Line 755, what does it mean by small buttons of growth?
Line 757, >80% reduction, the MIC is noted as >90% reduction, why the author indicated 80%
Where are the curves?
What OD did they use to measure the MIC?
What are the volumes used for both inoculum and the compounds?
What is the positive control used for Actinomyces viscosus?
Author Response
Dear Editors,
Dear Reviewer #1:
Dear Editors, I am very grateful for the helpful, which you doing for the authors. We have had plenty of mistakes, indicated with the help of the Referees. Taking into account the wishes and comments of the Reviewers of our article, we made important corrections and additions to the manuscript that were necessary for the better presentation of our scientific materia
Authors were very grateful for the valuable remarks from Referee 1.
Comments to the Author:
Line 51, authors mentioned phthalimide, although there is no phthalimide in figure1.
Figure 1, authors need to show what is R, R1 and R2
We have modified Figure 1, and shown the substituents
In Scheme 1, compound 1 was not used, why its structure there?
Compound 1 was the started compounds in this work. Compounds 2 and also the azide 35 were synthesized from 1. We slightly modified Scheme 1.
Line 81, azide and not aside
Thank you very much
Line 93, coumarin 6 and not 5.
Thank you very much
All bacterial names should be italic and they need to include the ATCC numbers
This error has been corrected (See experimental Section, 4.3.1).
In the work was we used strain Staphylococcus aureus 209р ATCC 6538-P and the strains from the collection of Department of Microbiology, Immunology and Virology, Novosibirsk State Medical University: Staphylococcus aureus С-18 - clinical isolated strain, Staphylococcus aureus "Viotko" and Actinomyces viscosus U-18.
Authors didn't include standard controls
As a positive control of antibacterial activity, the drug Ceftriaxone was used, which in the given conditions of the experiment showed the values of MIC equal:
for Staphylococcus aureus 209р ATCC 6538-P - 0,97±0, 02 µg/ml;
for Staphylococcus aureus С-18 – 6,50±1,29 µg/ml;
for Staphylococcus aureus "Viotko" - 1,03±0,32 µg/ml;
for Actinomyces viscosus U-18 933,3±33,33 µg/ml.
However, the main focus at the present stage of the work was to identify potential antibacterial activity and compare it within the studied group of substances and choose among them the most active for further study.
What concentrations authors used in the antibacterial assay?
To assess antibacterial activity, a number of double dilutions of the studied substances starting from 1000 µg/ml were used. (See experimental Section, 4.3.1; 4.3.2).
Where is the negative control?
As a negative control, the test culture was introduced into 1 ml of broth and cultivated under the same conditions, followed by sowing on an agar nutrient medium and taking into account the growth of bacteria.
Why did authors use E.coli JM109?
This strain is most similar to wild type of E. coli in our bacterial collection.
Authors showed the concentrations in uM and the MIC in ug/mL..how does this work?
This error has been corrected (Exp. Sect., 4.3.2).
Line 755, what does it mean by small buttons of growth?
This was a mistake
Line 757, >80% reduction, the MIC is noted as >90% reduction, why the author indicated 80%
This was a mistake
Where are the curves?
We have included the curves in the Suppl material.
What OD did they use to measure the MIC?
In the method of macro-dilutions the standardization of inoculum preparation was carried out using turbidimetry, but was controlled directly by seeding the final dilution of the suspension to the surface of the agar medium with further calculation of the number of colony-forming units (CFU).
What are the volumes used for both inoculum and the compounds?
The experiment was carried out in the total volume of the nutrient medium equal to 1.0 ml, where the studied substance was introduced in the desired dose and the test culture of the microorganism in an amount (2.75±0.85)103 CFU.
What is the positive control used for Actinomyces viscosus?
A positive control against Actinomyces viscosus is its sensitivity to Ceftriaxone in dose 933,3±33,33 µg/ml.
Thank you very much. Typos and errors have been corrected.
We hope the revised manuscript can be accepted for publication in Molecules.
Yours sincerely,
Elvira Shults
Reviewer 2 Report
The manuscript deals with the isolation and investigation into antibacterial activity of a series of triazole-derived coumarin hybrids. Overall, the article is original and interesting. Its content is well organized, experiments are well designed and concluded In a proper manner.
Unfortunately, several grammar inconsistencies and phrasing mistakes are present in the whole text. These issues cause a lowering in the scientific soundness of the article and in some cases are quite confusing.
Examples:
Line 37 – I think the sentence lacks correct phrase. I would assume that the authors refer to “broad array of biological activity” rather than “broad spectrum” which on its own suggests the reference to spectroscopy. Please check.
Paragraph 2 of the manuscript and particularly lines 39-44 require grammar check and some additional corrections. (e.g Line 39 – Since authors refer to various structurally different coumarins the word “Various” seems more appropriate rather than the word “Different”.)
Several grammar inconsistencies and phrasing issues need correction throughout the text (especially the introduction section).
These, and several other language issues should be addressed prior the final acceptance of the manuscript.
Therefore I strongly recommend the professional academic proofreading of the manuscript by native speaker.
Author Response
Dear Editors,
Dear Reviewer #2:
Dear Editors, I am very grateful for the helpful, which you doing for the authors. We have had plenty of mistakes, indicated with the help of the Referees. Taking into account the wishes and comments of the Reviewers of our article, we made important corrections and additions to the manuscript that were necessary for the better presentation of our scientific materia
Authors were very grateful for the valuable remarks from Referee 2.
Comments to the Author:
Unfortunately, several grammar inconsistencies and phrasing mistakes are present in the whole text. These issues cause a lowering in the scientific soundness of the article and in some cases are quite confusing.
Examples:
Line 37 – I think the sentence lacks correct phrase. I would assume that the authors refer to “broad array of biological activity” rather than “broad spectrum” which on its own suggests the reference to spectroscopy. Please check.
Paragraph 2 of the manuscript and particularly lines 39-44 require grammar check and some additional corrections. (e.g Line 39 – Since authors refer to various structurally different coumarins the word “Various” seems more appropriate rather than the word “Different”.)
Several grammar inconsistencies and phrasing issues need correction throughout the text (especially the introduction section).
These, and several other language issues should be addressed prior the final acceptance of the manuscript.
Therefore I strongly recommend the professional academic proofreading of the manuscript by native speaker.
Thank you very much. Typos and errors have been corrected.
We hope the revised manuscript can be accepted for publication in Molecules.
Yours sincerely,
Elvira Shults
Reviewer 3 Report
The manuscript by Elvira E. Shults and co-workers described design and synthesis of coumarin-1,2,3-triazole hybrids obtained from natural furocoumarin peucedanin and evaluation of their antibacterial activity. The synthetic part of the manuscript consists of a large number of transformations and products. The rational and the synthesis are well described and the experiments appear to be executed with care. Some of the products showed excellent antibacterial activity against S.aureus strains, with sum micro molar MIC values. Compound 37c was found to be selective against Bacillius subtilis and E.coli with MIC value 0.02-0.15 μg/mL. The manuscript certainly deserves publication in Molecules, however, here are some minor issues that need to be addressed.
13C NMR resonances are listed to two decimal places, which is rather unconventional. More importantly, all proton and carbon resonances are assigned. As there is no indication in the manuscript that the assignment was done on the basis of 2D NMR experiments, it is recommended to be removed. It may be dangerous to assign resonances without 2D spectra, as it can lead to unnecessary misinterpretations.
Author Response
Dear Editors,
Dear Reviewer #3:
Dear Editors, I am very grateful for the helpful, which you doing for the authors. We have had plenty of mistakes, indicated with the help of the Referees. Taking into account the wishes and comments of the Reviewers of our article, we made important corrections and additions to the manuscript that were necessary for the better presentation of our scientific materia
Comments to the Author:
13C NMR resonances are listed to two decimal places, which is rather unconventional. More importantly, all proton and carbon resonances are assigned. As there is no indication in the manuscript that the assignment was done on the basis of 2D NMR experiments, it is recommended to be removed. It may be dangerous to assign resonances without 2D spectra, as it can lead to unnecessary misinterpretations.
Authors were very grateful for the valuable remarks from Referee 3.
The assignment of protons was done on the base of analysis of the spectra of the new compounds compared with the spectra of the starting compounds, NMR signal assignments of which were carried out with the aid of a combination of 1D and 2D NMR techniques that included 1H, 13C, COSY, heteronuclear Single Quantum Correlation (HSQC) and Heteronuclear Multiple Bond Correlation (HMBC).
The carbon resonance spectra of all newly synthesized compounds were corrected and given without assigmment.
Thank you very much again.
We hope the revised manuscript can be accepted for publication in Molecules.
Yours sincerely,
Elvira
Round 2
Reviewer 1 Report
Authors fulfilled all the required comments and modified the manuscript accordingly.